# INDEPENDENTLY-PREPARED QUERY-EFFICIENT MODEL SELECTION

## ABSTRACT

The advancement of deep learning technologies is bringing new models by the day, which not only facilitates the importance of model selection but also makes it more challenging than ever. However, existing solutions for model selection either require a large amount of model operations proportional to the number of candidates when selecting models for each task, or require group preparations that jointly optimize the embedding vectors of many candidate models. As a result, the scalability of existing solutions is limited with the increasing amounts of candidates. In this work, we present a new paradigm for model selection, namely independently-prepared query-efficient model selection. The advantage of our paradigm is twofold: first, it is query-efficient, meaning that it requires only a constant amount of model operations every time it selects models for a new task; second, it is independently-prepared, meaning that any information about a candidate model that is necessary for the selection can be prepared independently requiring no interaction with others. Consequently, the new paradigm offers by definition many desirable properties for applications: updatability, decentralizability, flexibility, and certain preservation of both candidate privacy and query privacy. With the benefits uncovered, we present Standardized Embedder as a proof-of-concept solution to support the practicality of the proposed paradigm. We empirically evaluate this solution by selecting models for multiple downstream tasks, from a pool of 100 pre-trained models that cover different model architectures and various training recipes, highlighting the potential of the proposed paradigm.

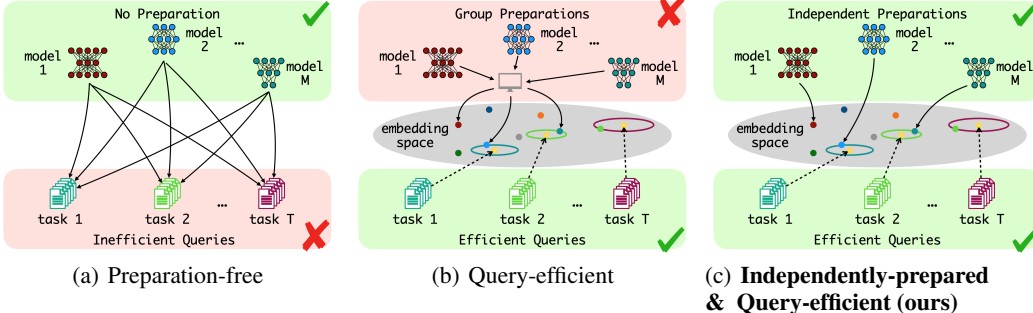

Figure 1: Illustrations of different paradigms for model selection. (a) Preparation-free Model Selection: Every candidate model needs to be directly examined for each query. (b) Query-efficient Model Selection: Embedding vectors of the candidates are jointly prepared in advance and models are selected based on vector comparisons. **(c) Independently-prepared Query-efficient Model Selection (ours)**: The embedding vectors can be independently prepared for different candidates and models are selected based on vector comparisons.

# 1 INTRODUCTION

New models are being created and becoming available at a rate beyond previous imaginations. Hugging Face Hub[1], a web platform for hosting machine learning models, datasets, and demo applications, included more than 300k pre-trained models in August 2023, when its owner, the company named Hugging Face, obtained a $4.5 billion valuation while raising a funding of $235 million backed by Salesforce, Google, Nvidia, Intel, AMD, Qualcomm, IBM, Amazon and more[2]. The vast amounts of new models can be highly valuable assets if we can identify the ones suitable for tasks of interest. Nevertheless, as the number of candidates grows, selecting proper models for a new task becomes increasingly challenging. *How can we make model selection truly scalable to meet the demands now and in the future?*

**Existing Paradigm 1: Preparation-free Model Selection.** The most naive solution for model selection is to try every candidate model on the task of interest. This can select the best possible candidates but becomes prohibitively expensive when dealing with more than a handful of models. A previous direction towards addressing this issue is the study of transferability metrics for models (Bao et al., 2019; Nguyen et al., 2020; You et al., 2021; Pándy et al., 2022; Huang et al., 2022; Agostinelli et al., 2022; Bolya et al., 2021). Transferability metrics are scores that can be computed without training models (typically requiring only forward passes of models) and correlate with the performance of models after being transferred to downstream tasks. By computing such metrics for candidate models instead of training them, one can greatly reduce the computational overhead by eliminating the cost of training multiple models. We refer to them as preparation-free model selection since they require no additional preparation before queries (i.e. selecting models for downstream tasks), as illustrated in Figure 1(a). However, despite the enhanced efficiency with transferability metrics, the number of **model operations** (e.g. forward/backward passes of models and uploads/downloads of model weights) required per query still depends linearly on the number of candidates in this paradigm. Thus it can remain costly as the number of candidates increases.

**Existing Paradigm 2: Query-efficient Model Selection.** As a gift from the study of task similarities (Zamir et al., 2018; Achille et al., 2019; Liu et al., 2022a; Zhou et al., 2022), Model2Vec (Achille et al., 2019) offers a solution with a more desirable query complexity. Informally, each candidate model is associated with an embedding vector, and these embedding vectors are trained jointly to predict the best candidates from their distances to the task embedding. After this, for each query, the model selection is reduced to the comparisons of embedding vectors, which requires only vector operations that are orders of magnitude faster than forward/backward passes of typical neural models (i.e. model operations). We refer to this paradigm as query-efficient model selection since the number of model operations required for each query is O(1) with respect to the number of candidates (which is spent to compute the embedding of the query task), as illustrated in Figure 1(b). However, query-efficient model selection suffers from another scalability issue that originates from its preparations: updatability. Unlike preparation-free model selection schemes such as transferability metrics, Model2Vec (Achille et al., 2019) requires a group preparation stage where the embedding vectors of all candidate models are learned jointly so that their distances to task embedding vectors correlate with their performance. However, such group preparations can be increasingly costly and difficult to update as new candidate models continue to join.

**New Paradigm: Independently-prepared Query-efficient Model Selection.** In this work, we propose a new paradigm, namely **independently-prepared query-efficient model selection**. In addition to being query-efficient by comparing embedding vectors as in the previous paradigm, the new paradigm requires the solutions to be independently-prepared, meaning that each candidate model can be mapped to its embedding vectors independently, without relying on any interaction with other candidate models, as illustrated in Figure 1(c). Such property improves the scalability of query-efficient model selection not only by enabling trivial distribution of computations to multiple machines and easy updates with new candidate models, but also by allowing model owners to freely enroll their models as candidates of selection by themselves without directly revealing their models to the public: They simply prepare and publish the embedding vectors corresponding to their own models. Users who want to select models for downstream tasks can therefore collect the embedding vectors from potentially different sources and select models efficiently based on vector comparisons.

---

[1]https://huggingface.co
[2]https://www.nasdaq.com/articles/ai-startup-hugging-face-valued-at-$4.5-bln-in-latest-round-of-funding

Eventually, only the owners of the selected candidates need to be notified for access to the models while all other model owners do not reveal their models and are not even aware of the selections. **To summarize**, attributing to the nature of independent preparations, the proposed paradigm offers desirable properties including **updatability**, **decentralizability**, **flexibility**, and **preservation of both candidate privacy and query privacy**, which will be elaborated in detail in Section 3.2.

*How to design solutions for independently-prepared query-efficient model selection?* In this work, we provide a proof-of-concept design called Standardized Embedder. The key intuition of Standardized Embedder is standardization, i.e. using one public model as the baseline to embed all different candidate models, thus ensuring the independently learned embedding vectors conform to the same standard and are therefore comparable. We empirically evaluate the effectiveness of Standardized Embedder to select from a pool of 100 models for multiple downstream tasks, highlighting the practicality and potentials of the proposed paradigm.

In summary, our main contributions include:

- We propose **a new model selection paradigm**, namely independently-prepared query-efficient model selection, which offers by definition **desirable properties** including updatability, decentralizability, flexibility, and preservation of both candidate and query privacy;

- We present Standardized Embedder as a **proof-of-concept solution** in the proposed paradigm;

- We evaluate the solution to highlight the potential of the proposed paradigm by **gathering 100 pre-trained models**, which cover different architectures and training recipes, and using them as candidates of model selection for different downstream tasks.

## 2 RELATED WORK

**Preparation-free Model Selection: Transferability Metrics.** Intuitively, transferability metrics are scores that correlate with the performance of models/features after being transferred to a new task and can be computed without training: H-score (Bao et al., 2019) is defined by incorporating the estimated inter-class variance and the redundancy of features; LEEP (Nguyen et al., 2020) estimates the distribution of target task label conditioned on the label of pre-trained tasks to construct a downstream classifier without training and use its performance as the metric; LogME (You et al., 2021) estimates the maximum value of the marginalized likelihood of the label given pre-trained features and uses its logarithm as the score; GBC (Pándy et al., 2022) uses class-wise Gaussians to approximate downstream samples in the pre-trained feature space so that class overlaps can be computed with Bhattacharyya coefficients to serve as a score; TransRate (Huang et al., 2022) estimates mutual information between features and labels by resorting to coding rate. Separate evaluations of transferability metrics are conducted by Agostinelli et al. (2022) and Bolya et al. (2021).

**Query-efficient Model Selection: Model2Vec.** Model2Vec is proposed jointly with Task2Vec by Achille et al. (2019). The goal of Task2Vec is to embed different tasks into a shared vector space and it does so by estimating the Fisher information matrix with respect to a probe network. To embed models, they first initialize the embedding of each model as the sum of the Task2Vec embedding $F$ of its pre-trained task (which is set to 0 if the task is unknown) and a learnable perturbation $b$. Then they learn the perturbations of all models jointly by predicting the best model given the distances of model embeddings to the task embeddings. This design, while intuitive, leads to updatability issues when embedding new models. In addition, access to multiple downstream tasks is required in the preparation stage, which further limits its applicability.

## 3 INDEPENDENTLY-PREPARED QUERY-EFFICIENT MODEL SELECTION

### 3.1 FORMAL DESCRIPTION

Here we present the formal description of independently-prepared query-efficient model selection, which can be divided into two parts: (1) **preparations**, where candidate models are converted into embedding vectors for later use in selecting models; (2) **queries**, where models are selected given downstream data (i.e. data corresponding to the task of interest) and the model embeddings generated during preparations.

**Preparations.** The preparation scheme is defined by a single function of the form $\mathcal{P}(\text{model}) \to \mathcal{V}$, where 'model' denotes a single (candidate) model and $\mathcal{V}$ denotes an embedding space. We refer to $\mathcal{P}$ as the preparation function. Intuitively, $\mathcal{P}$ maps a single model into its corresponding embedding vectors. Let $f_1, f_2, \ldots, f_M$ be all candidate models where $M$ is the total number of models. In this stage, embedding vectors of all candidate models are generated as $\{v_i = \mathcal{P}(f_i)\}_{i=1}^M$, where $v_i$ is the embedding vector corresponding to candidate model $f_i$. The preparations are **independent** in this paradigm in the sense that the preparation function $\mathcal{P}$ takes a single candidate model as its input and is applied independently to each candidate model.

**Queries.** Queries are defined by a single function of the form $\mathcal{Q}(\text{data}, \text{model embeddings}) \to$ model indices, where 'data' denotes the downstream data corresponding to the task of interest and 'model embeddings' denote the embeddings generated during preparations. We refer to $\mathcal{Q}$ as the query function. Intuitively, $\mathcal{Q}$ selects one or more candidate models based on both downstream data corresponding to the task of interest and the model embeddings generated during preparations. The queries are **efficient** in the sense that the selection of models is reduced to the selection of vectors with model embeddings and task embeddings: In this way, the number of model operations (e.g. forward/backward passes) spent is O(1) with respect to the number of candidates (i.e. the complexity of model operations does not grow as the number of candidates increases).

## 3.2 IMPLICATIONS

Intuitively, the proposed paradigm is obtained by combining independent preparations with the query-efficient model selection paradigm. With independent preparations, the proposed paradigm offers by definition many desirable properties: **updatability**, **decentralizability**, **flexibility**, and certain **preservation of candidate privacy**, in addition to certain **preservation of query privacy** inherited from query-efficient model selection paradigm.

**Updatability:** The paradigm is updatable as the cost of adding new candidate models does not increase when the total number of candidates increases. Since the preparations are independent, the only necessary update when a new candidate model joins is to apply the preparation function $\mathcal{P}$ to the new model for the corresponding model embedding. No modification is required to any previously generated model embeddings.

**Decentralizability:** The selection process is naturally decentralizable as model owners can enroll their models as candidates of future model selections entirely on their own, requiring no collaboration with other model owners and no centralized coordination. Given the preparation function $\mathcal{P}$, model owners can independently embed their models and publish/broadcast the resulting embeddings by themselves. After that, any party with access to the published model embeddings can use the query function $\mathcal{Q}$ to select models based on the data for its own task.

**Flexibility:** The paradigm is flexible as it can easily support model selection across different implementation frameworks, different owners, and different platforms. Firstly, attributing to the independent preparations, different owners and platforms can embed models independently with the preparation function $\mathcal{P}$, potentially implemented in different frameworks. Secondly, since the only candidate-related inputs to the query function $\mathcal{Q}$ are the model embeddings, the only remaining barrier is to collect them. Fortunately, model embeddings are typically very portable as they are just real vectors, which can be stored and processed in diverse formats through various tools. In addition, typical dimensions of model embeddings are fairly small: For instance, in later empirical evaluations, we incorporate settings with 512-dimensional and 768-dimensional model embeddings, while as an informal comparison, each of Figure 1(a), 1(b) and 1(c) contains $1510 \times 1170 \approx 1.8 \times 10^6$ RGB pixels, which is about $5.3 \times 10^6$ dimensions.

**Preservation of candidate privacy:** The process preserves the privacy (and/or intellectual properties) of model owners as no model-related information other than their embeddings is released to any other parties (e.g. other model owners and the owners of the target tasks) throughout the selection.

**Preservation of query privacy:** This framework preserves the privacy of users who select models for their tasks as not only will their task data be kept private but also model owners will have no knowledge regarding if there is a selection or not, unless the users notify them. After all, once all model embeddings are collected, users can finish the selection entirely locally.

Table 1: Properties of different model selection paradigms.

| model selection paradigm | scalable preparation | efficient query | updatable | decentralizable | flexible | candidate privacy | query privacy |
|---|---|---|---|---|---|---|---|
| preparation-free | ✓ | | ✓ | ✓ | | | |
| query-efficient | | ✓ | | | | | ✓ |
| **ours** | ✓ | ✓ | ✓ | ✓ | ✓ | ✓ | ✓ |

We conclude this section with the above table summarizing properties of model selection paradigms. Notably, the proposed paradigm is potentially applicable to different scenarios relying on different subsets of the properties: In centralized settings where a single party has access to not only all candidate models but also the data of the target task, the updatability can be the most important; In hybrid settings where the candidate models are hosted by one party and the task data is hosted by another, the preservation of candidate privacy and query privacy also become critical; In decentralized settings where candidate models are possessed by multiple parties, all properties are valuable.

## 4    A PROOF-OF-CONCEPT SOLUTION: STANDARDIZED EMBEDDER

In this section, we introduce the design of Standardized Embedder as a proof-of-concept solution to independently-prepared query-efficient model selection. We view every deep model as a collection of features, where each feature is a function mapping the input space to scalar values. The key intuition of our design is to define computationally tractable vectors characterizing the representation powers of a set of features with respect to the baseline features, so that these individually and independently extracted vectors can be comparable for model selections.

As illustrated in Figure 1(c), this baseline solution contains two major components: **independent preparations**, where model embedding vectors are learned independently for different candidates, and **efficient query**, where a task embedding vector is learned from the downstream data and is used to search among model embeddings to guide model selection. These two components are presented respectively in Section 4.2 and 4.3, built on concepts we presented in Section 4.1.

### 4.1    TOOL: (APPROXIMATE) FUNCTIONALITY EQUIVALENCE FOR SETS OF FEATURES

Firstly, we introduce notations. Let $\mathcal{X}$ be the input space. A feature $f : \mathcal{X} \to \mathbb{R}$ is defined as a function mapping any sample from the input space $\mathcal{X}$ to a real number. A set of features $F$ is therefore a set of functions, which can also be considered as a function $F : \mathcal{X} \to \mathbb{R}^n$ mapping any sample to a vector of $n$ dimensions, where $n$ can be either finite or countably infinite, depending on whether or not the set contains a finite number of features.

**Definition 1** (Functionality Equivalence). *For two sets $F : \mathcal{X} \to \mathbb{R}^n$ and $\hat{F} : \mathcal{X} \to \mathbb{R}^m$ of features, they are considered $\delta$-equivalent in functionality over a distribution $D$ over $\mathcal{X}$, if and only if there exist two affine transformations $w, b \in \mathbb{R}^{n \times m} \times \mathbb{R}^m$ and $\hat{w}, \hat{b} \in \mathbb{R}^{m \times n} \times \mathbb{R}^n$ such that*

$$\begin{cases} \mathbb{E}_{x \sim D} \left[ S_{cos} \left( w^\top F(x) + b, \hat{F}(x) \right) \right] \geq 1 - \delta \\ \mathbb{E}_{x \sim D} \left[ S_{cos} \left( F(x), \hat{w}^\top \hat{F}(x) + \hat{b} \right) \right] \geq 1 - \delta \end{cases}$$

*where $S_{cos}(u, v)$ denotes cosine similarity between two vectors $u$ and $v$.*

Functionality equivalence characterizes cases where two sets of features are considered the same regarding their usability in unknown applications. Intuitively, since most (if not all) modern architectures of neural networks have at least one affine transformation following the feature layers, two sets of features should be considered equivalent even if they differ by an affine transformation. Similar arguments are introduced by Wang et al. (2018) to understand deep representations, where they consider two representations to be equivalent when the subspaces spanned by their activation vectors are identical. While in principle other similarity metrics can be utilized as well, we use cosine similarity in this work since it is naturally invariant to feature scalings.

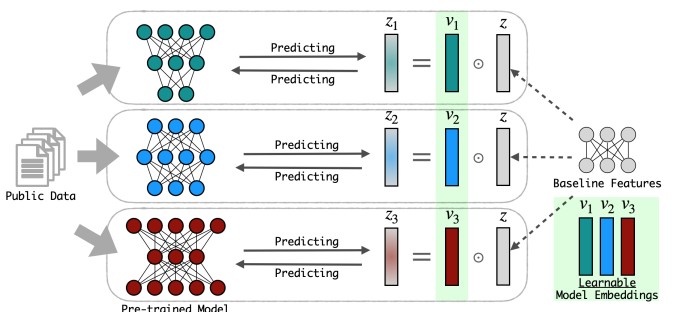 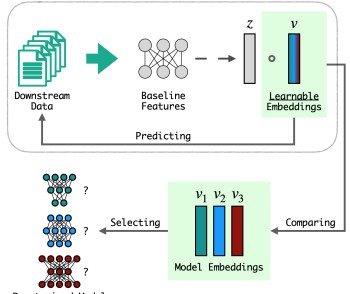

(a) Preparation: independent embedding of models     (b) Query: embedding of tasks

Figure 2: An illustration of Standardized Embedder. (a) Using features of a public model as the baseline, a vector embedding is learned **independently** for each pre-trained model. The embeddings of models denote their approximately equivalent feature subsets in the baseline features. (b) Task embeddings are defined by subsets of the baseline features that are important to corresponding downstream tasks, which are identified through enforcing sparsity regularization.

## 4.2 PREPARATION: INDEPENDENTLY LEARNED MODEL EMBEDDINGS THROUGH IDENTIFYING EQUIVALENT FEATURE SUBSETS

With functionality equivalence from Definition 1, we can characterize the representation powers of any set of features by associating them with the equivalent subsets from a pre-defined, baseline feature set $B : \mathcal{X} \to \mathbb{R}^N$ (Empirically we will use a public model as this baseline feature set, which will be elaborated in Section 5). Since any subset of the baseline feature set $B : \mathcal{X} \to \mathbb{R}^N$ can be directly associated with a binary vector from $\{0,1\}^N$ (i.e. each 1 indicating the presence of a feature and each 0 indicating an absence; See Appendix C for examples), we simply use such vectors as the embeddings of models. Notably, while technically we define binary embedding vectors here, the embedding space will be relaxed to a continuous one (i.e. $[0,1]^N$) for the actual algorithm. The formal definition is as follows.

**Definition 2** (Vector Embedding through Equivalent Feature Subsets). *Given a baseline feature set $B : \mathcal{X} \to \mathbb{R}^N$, a vector $v \in \{0,1\}^N$ is a $\delta$-embedding vector of a feature set $F : \mathcal{X} \to \mathbb{R}^n$ over a distribution $D$ if and only if $F$ and $\{B_i | v_i = 1\}$ are $\delta$-equivalent in functionality over $D$.*

Given a set of features as the baseline, the embedding vectors corresponding to a set of features are defined through Definition 2. Consequently, we can now conceptually map each model, represented as a set of features, into a vector embedding space that associates with the baseline features.

In practice, to compute the embedding vectors given baseline features, we relax the binary embedding space to a continuous one and reformulate it as the following optimization:

$$\max_{v,w,b,\hat{w},\hat{b}} \min\left(L_{\text{to baseline}}, L_{\text{from baseline}}\right)$$

$$s.t.\ L_{\text{to baseline}} = \mathbb{E}_{x \sim D}\left[S_{\cos}\left(w^\top F(x) + b, v \odot B(x)\right)\right]$$

$$L_{\text{from baseline}} = \mathbb{E}_{x \sim D}\left[S_{\cos}\left(F(x), \hat{w}^\top (v \odot B(x)) + \hat{b}\right)\right]$$

$$v \in [0,1]^n \text{ and } w, b \in \mathbb{R}^{n \times N} \times \mathbb{R}^N, \hat{w}, \hat{b} \in \mathbb{R}^{N \times n} \times \mathbb{R}^n$$

where $F : \mathcal{X} \to \mathbb{R}^n$ is the feature set that we want to vectorize, $B : \mathcal{X} \to \mathbb{R}^N$ is the set of baseline features, $D$ is the underlying data distribution, $v$ is the (relaxed) embedding vector, $w, b, \hat{w}, \hat{b}$ are parameters of affine transforms and $\odot$ denotes Hadamard product (i.e. element-wise multiplication). An illustration of the model embedding process is included in Figure 2(a).

Empirically, the constraint $v \in [0,1]^n$ is implemented via reparameterization through the sigmoid function, i.e. $v_i = 1/(1 + e^{-v_i'/\tau})$, where $\tau$ is a constant known as temperature and we use a fixed temperature of $\tau = 0.01$ in all experiments. Intuitively, the optimization wants to find a subset of the baseline features (indicated by the mask $v$) that is $\delta$-equivalent to $F$ for smaller $\delta$.

Both Definition 2 and the relaxation are straightforward, but it is worth noting that the embedding depends on not only the model (i.e. the set of features) to be embedded, but also the set of baseline features, and the embedding vectors may not be unique by definition depending on the choice of baseline features. Conceptually, what we do here is to compare the embedding distributions of different models by drawing a single embedding vector from each distribution.

## 4.3 QUERY: TASK EMBEDDINGS THROUGH FEATURE SIFTING

With models embedded in a vector space, the missing piece to enable model selections through vector comparisons is to identify a reference point (e.g. a task embedding) in the embedding space from the context of downstream applications. In this section, we showcase how one can derive the task embedding vector from downstream data and how to select competitive models by comparing only embedding vectors. An illustration of the process is included in Figure 2(b): Intuitively, we derive a reference point in the vector space by identifying subsets of the baseline features $B : \mathcal{X} \to \mathbb{R}^N$ that are important to the task of interest, which can then be directly associated with binary vectors from $\{0, 1\}^N$, similar to how we previously embed models as vectors.

Formally, for a downstream task, let $\mathcal{X}$ be the input space, $\mathcal{Y}$ be the label space, we use $\hat{D}$ to denote the downstream data distribution, which is a distribution over $\mathcal{X} \times \mathcal{Y}$. Using $L$ to denote the corresponding task loss, identifying important features can be formulated as follows:

$$\min_{v, w, b} \; \mathbb{E}_{x, y \sim \hat{D}} \left[ L \left( w^\top (v \odot B(x)) + b, y \right) \right] + \gamma ||v||_1$$
$$s.t. \; ||w^\top||_1 = 1 \text{ and } v \in [0, 1]^n$$

where $v \in [0, 1]^n$ is the embedding vector of the task to be learned, $w, b \in \mathbb{R}^{n \times |\mathcal{Y}|} \times \mathbb{R}^{|\mathcal{Y}|}$ jointly denotes a prediction head associated with the task of interest, $||v||_1$ denotes the $\ell_1$ norm of the embedding vector (which functions as sparsity regularization), $||w^\top||_1$ denotes the matrix norm of $w^\top$ induced by $\ell_1$ norm of vectors (i.e. $||w^\top||_1 = \sup_{x \neq 0} ||w^\top x||_1 / ||x||_1 = \max_i \sum_j |w_{ij}|$) and $\gamma$ is sparsity level, a scalar hyper-parameter controlling the strength of the sparsity regularization $\gamma ||v||_1$. A rule of thumb for choosing $\gamma$ is discussed in Section 5.4.

**Selection metric.** Given the task embedding, we compare it with embeddings of candidate models to identify the most similar ones to the task embedding with respect to a similarity metric—-the corresponding models are the ones to be selected. Notably, all our embedding vectors, including model embeddings and task embeddings, are relaxations of binary vectors denoting subsets of the baseline features. This is well related to fuzzy set theory (Zadeh, 1965; Klir & Yuan, 1995) where each element can have a degree of membership between 0 and 1 to indicate whether it is not/fully/partially included in a set. Interpreting both model embeddings and task embeddings as fuzzy sets, we incorporate standard fuzzy set intersection to measure the similarity between model embeddings and task embeddings. Formally, let $u, v \in [0, 1]^n$ be two embedding vectors (interpreted as fuzzy sets), the cardinality of their standard intersection is simply $I_{\text{standard}}(u, v) = \sum_{i=1}^n \min(u_i, v_i)$.

Intuitively, the task embedding denotes the set of baseline features useful for the task of interest and each model embedding denotes the set of baseline features possessed by the corresponding candidate model. Thus informally the cardinality of their intersection measures the quantity of the useful features owned by candidate models and therefore provides guidance for downstream performance.

# 5 EVALUATION

## 5.1 EVALUATION SETUP

For empirical evaluations, we gather **100 pre-trained models** with various architectures and training recipes as the candidates of model selection. The details are included in Appendix A.

**Preparations.** Recalling that Standardized Embedder uses a public model as the baseline features and (approximate) functionality equivalence is defined over an underlying distribution D, we include different choices of baseline features, a pre-trained ResNet-18 or a pre-trained Swin Trans-

former (tiny) and we use the validation set of ImageNet (50000 samples in total) as the underlying distribution D (for the embedding of candidate models). We use the same hyper-parameters when embedding different candidate models: We use SGD optimizer with a batch size of 128, an initial learning rate of 0.1, a momentum of 0.9, and a weight decay of 5e-4; We divide the learning rate by a factor of 10 every 1k steps for settings with 4k training steps (=10.24 epochs) per candidate, and every 3k steps for settings with 10k training steps (=25.6 epochs) per candidate.

**Queries.** We evaluate the ability of Standardized Embedder to select competitive models for three downstream benchmarks, CIFAR-10 (Krizhevsky et al., 2009), CIFAR-100 (Krizhevsky et al., 2009) and STL-10 (Coates et al., 2011), which are natural object classifications benchmarks with varying granularity and varying domain shifts (compared to ImageNet validation set that we use as the distribution to embed models). To learn task embeddings, we use SGD optimizer with a batch size of 128, an initial learning rate of 0.1, a momentum of 0.9, and a weight decay of 5e-4 for 60 epochs, with the learning rate divided by a factor of 10 every 15 epochs. Notably, the weight decay is disabled for the task embedding to be learned to avoid interfering with the sparsity regularization from Section 4.3. For coherence, we defer to Section 5.4 the choice of the sparsity level $\gamma$.

**Ground truth.** To evaluate model selection, one has to collect the actual performance of every candidate model on each downstream task (Note that this is not really scalable as it is essentially the naive selection scheme that tries every candidate out, thus limiting the scales of our evaluation). Here for simplicity, we incorporate a linear probing protocol for evaluation: For every candidate model and for each downstream task, we train a linear classifier over its features with the SGD optimizer as the downstream accuracy (of this candidate model with respect to this downstream task), using a batch size of 128, an initial learning rate of 0.1, a momentum of 0.9, and a weight decay of 5e-4 for 60 epochs, with the learning rate divided by a factor of 10 every 15 epochs.

## 5.2 THE PERFORMANCE OF STANDARDIZED EMBEDDER

In Table 2, we include the quantitative evaluations on the performance of Standardized Embedder. For each downstream task, we report on the left half of the table the downstream accuracy of the best candidates (i.e. the ground truth) for references and report on the right half of the table the downstream accuracy of models selected (i.e. the top-1/top-3/top-5 candidates according to the selection metric). Standardized Embedder successfully locates models comparable to the best possible candidates with respect to different downstream tasks: When selecting only 1 model from the 100 candidates, the **worst** accuracy gap **across all evaluated settings** of Standardized Embedder (i.e. with different baseline features and training steps per candidate) and all downstream tasks evaluated is 3.34%, which is reduced to 2.45% when selecting 3 models and 1.47% when selecting 5 models; For the **best evaluated setting** of Standardized Embedder, which uses Swin Transformer (tiny) as baseline features and 10k steps per candidate, the gap is **at most** 1.10% for all evaluated downstream tasks even when selecting only 1 model.

In Figure 3 and Figure 4, we present both the downstream accuracy and the cardinality of standard intersection (i.e. the selection metric) for different downstream tasks when using different baseline features. We use the dashed line to highlight the downstream accuracy of the public model used as baseline features. An important observation here is that Standardized Embedder is able to locate much more competitive models when the public model used as baseline features is suboptimal.

## 5.3 ON THE CHOICE OF BASELINE FEATURES

Table 2 suggests that Standardized Embedder performs better when using Swin Transformer (tiny) as baseline features compared to when using ResNet-18. To further understand this, we compare the selection metric of candidate models on each of the evaluated downstream tasks when using different baseline features in Figure 5 and Figure 6, where in all cases there are clusters of green/orange points in the bottom right, which corresponds to Transformers and hybrid models (i.e. models with both convolution and attention) that are overrated when using ResNet-18 as baseline features.

This is potentially attributed to the limitations of ConvNets (i.e. Convolutional Neural Networks): The size of their receptive fields prevents them from capturing some long-range correlations utilized by models with attention (i.e. Transformers and hybrid models). While this suggests models with attention, such as Swin Transformer (tiny) we evaluated, can be potentially better choices of baseline

Table 2: Empirical evaluations of Standardized Embedder with **100 pre-trained models** as the candidates (See Appendix A for the full list). Standardized Embedder successfully locates models comparable to the best candidates for corresponding downstream tasks.

| downstream task | best candidate (**ground truth**) | model used as baseline features | training steps per candidate | downstream accuracy of selected models (**+gap from the best candidate**) | | |
|---|---|---|---|---|---|---|
| | | | | best of top 1 | best of top 3 | best of top 5 |
| CIFAR-10 | 95.15% | ResNet-18 | 4k | 91.81% (**3.34%**) | 94.57% (**0.58%**) | 94.57% (**0.58%**) |
| | | | 10k | 94.36% (**0.79%**) | 95.12% (**0.03%**) | 95.12% (**0.03%**) |
| | | Swin-T (tiny) | 4k | 94.57% (**0.58%**) | 94.57% (**0.58%**) | 95.12% (**0.03%**) |
| | | | 10k | 95.12% (**0.03%**) | 95.12% (**0.03%**) | 95.12% (**0.03%**) |
| CIFAR-100 | 82.58% | ResNet-18 | 4k | 81.11% (**1.47%**) | 81.11% (**1.47%**) | 81.11% (**1.47%**) |
| | | | 10k | 80.13% (**2.45%**) | 80.13% (**2.45%**) | 81.57% (**1.01%**) |
| | | Swin-T (tiny) | 4k | 81.11% (**1.47%**) | 81.48% (**1.10%**) | 81.48% (**1.10%**) |
| | | | 10k | 81.48% (**1.10%**) | 81.48% (**1.10%**) | 81.48% (**1.10%**) |
| STL-10 | 99.24% | ResNet-18 | 4k | 98.76% (**0.47%**) | 98.76% (**0.47%**) | 98.76% (**0.47%**) |
| | | | 10k | 97.51% (**1.72%**) | 98.60% (**0.64%**) | 98.76% (**0.47%**) |
| | | Swin-T (tiny) | 4k | 97.69% (**1.55%**) | 98.60% (**0.64%**) | 98.60% (**0.64%**) |
| | | | 10k | 98.60% (**0.64%**) | 98.60% (**0.64%**) | 98.60% (**0.64%**) |

features than ConvNets, we want to emphasize that it remains open regarding what kind of models will serve best as baseline features for model selection purposes. For now, we conclude this section as follows: In settings evaluated, using Swin Transformer (tiny) as baseline features offers better selections of models, while the results from using ResNet-18 are already good.

## 5.4 On Choosing Sparsity Level in Embedding Downstream Tasks

In Section 4.3, we introduce a scalar hyper-parameter, the sparsity level $\gamma$, to control the strength of sparsity regularization $\gamma \|v\|_1$ when defining the embedding of the downstream task. Here we will present a rule of thumb that we use for choosing $\gamma$ empirically in our experiments.

Intuitively, the sparsity regularization works by penalizing the use of any feature and therefore only features that are critical enough for the downstream task will be utilized. As the sparsity level $\gamma$ increases, the subsets of features preserved will also be smaller. Informally, to determine the set of features necessary for the downstream task, one can keep increasing the sparsity level $\gamma$ until the downstream performance starts to drop. In Figure 7, we include downstream accuracy of the baseline features corresponding to varying sparsity level $\gamma$ to present a rule of thumb for deciding the values of sparsity level: simply using the smallest $\gamma$ with at least 3% accuracy drop from the converged accuracy (i.e. the eventual accuracy when the sparsity level keeps decreasing). This single rule is applied to all experiments and it works well as previously presented in Section 5.2.

## 6 Conclusion

In this work, we propose a new paradigm of model selection that offers by definition properties including updatability, decentralizability, flexibility, and preservation of both candidate and query privacy. We present Standardized Embedder as a proof-of-concept solution and evaluate its performance to select from a pool of 100 pre-trained models for multiple downstream benchmarks. Empirically, this solution successfully locates models comparable to the best possible candidates. Despite some limitations of our evaluation which are detailed in Appendix D, this highlights the potential of the proposed paradigm. Through these, we hope to facilitate future work that further leverages an ocean of pre-trained models, a rapidly increasing resource.

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

## A    FULL LIST OF CANDIDATE MODELS USED IN THE EXPERIMENTS

Table 3: A full list of the 100 pre-trained models that are used as the candidate models in the experiments.

| index | name (used by the corresponding source) | category | source | index | name (used by the corresponding source) | category | source |
|---|---|---|---|---|---|---|---|
| 1 | ResNet18_Weights.IMAGENET1K_V1 | ConvNet | torchvision | 51 | ShuffleNet_V2_X1_0_Weights.IMAGENET1K_V1 | ConvNet | torchvision |
| 2 | EfficientNet_B0_Weights.IMAGENET1K_V1 | ConvNet | torchvision | 52 | ShuffleNet_V2_X1_5_Weights.IMAGENET1K_V1 | ConvNet | torchvision |
| 3 | GoogLeNet_Weights.IMAGENET1K_V1 | ConvNet | torchvision | 53 | ShuffleNet_V2_X2_0_Weights.IMAGENET1K_V1 | ConvNet | torchvision |
| 4 | Swin_T_Weights.IMAGENET1K_V1 | Transformer | torchvision | 54 | Swin_V2_T_Weights.IMAGENET1K_V1 | Transformer | torchvision |
| 5 | MobileNet_V3_Large_Weights.IMAGENET1K_V1 | ConvNet | torchvision | 55 | ViT_B_32_Weights.IMAGENET1K_V1 | Transformer | torchvision |
| 6 | MobileNet_V3_Large_Weights.IMAGENET1K_V2 | ConvNet | torchvision | 56 | ViT_B_16_Weights.IMAGENET1K_V1 | Transformer | torchvision |
| 7 | MobileNet_V3_Small_Weights.IMAGENET1K_V1 | ConvNet | torchvision | 57 | ViT_B_16_Weights.IMAGENET1K_SWAG_LINEAR_V1 | Transformer | torchvision |
| 8 | MNASNet0_5_Weights.IMAGENET1K_V1 | ConvNet | torchvision | 58 | Wide_ResNet50_2_Weights.IMAGENET1K_V1 | ConvNet | torchvision |
| 9 | ShuffleNet_V2_X0_5_Weights.IMAGENET1K_V1 | ConvNet | torchvision | 59 | Wide_ResNet50_2_Weights.IMAGENET1K_V2 | ConvNet | torchvision |
| 10 | AlexNet_Weights.IMAGENET1K_V1 | ConvNet | torchvision | 60 | mobileone_s0 | ConvNet | timm |
| 11 | ConvNeXt_Tiny_Weights.IMAGENET1K_V1 | ConvNet | torchvision | 61 | mobileone_s1 | ConvNet | timm |
| 12 | ConvNeXt_Small_Weights.IMAGENET1K_V1 | ConvNet | torchvision | 62 | mobileone_s2 | ConvNet | timm |
| 13 | DenseNet121_Weights.IMAGENET1K_V1 | ConvNet | torchvision | 63 | mobileone_s3 | ConvNet | timm |
| 14 | DenseNet161_Weights.IMAGENET1K_V1 | ConvNet | torchvision | 64 | mobileone_s4 | ConvNet | timm |
| 15 | DenseNet169_Weights.IMAGENET1K_V1 | ConvNet | torchvision | 65 | inception_next_tiny.sail_in1k | ConvNet | timm |
| 16 | DenseNet201_Weights.IMAGENET1K_V1 | ConvNet | torchvision | 66 | inception_next_small.sail_in1k | ConvNet | timm |
| 17 | EfficientNet_B1_Weights.IMAGENET1K_V1 | ConvNet | torchvision | 67 | inception_next_base.sail_in1k | ConvNet | timm |
| 18 | EfficientNet_B2_Weights.IMAGENET1K_V1 | ConvNet | torchvision | 68 | ghostnet_100.in1k | ConvNet | timm |
| 19 | EfficientNet_B3_Weights.IMAGENET1K_V1 | ConvNet | torchvision | 69 | ghostnetv2_100.in1k | ConvNet | timm |
| 20 | EfficientNet_B4_Weights.IMAGENET1K_V1 | ConvNet | torchvision | 70 | ghostnetv2_130.in1k | ConvNet | timm |
| 21 | EfficientNet_V2_S_Weights.IMAGENET1K_V1 | ConvNet | torchvision | 71 | ghostnetv2_160.in1k | ConvNet | timm |
| 22 | Inception_V3_Weights.IMAGENET1K_V1 | ConvNet | torchvision | 72 | repghostnet_050.in1k | ConvNet | timm |
| 23 | MNASNet0_75_Weights.IMAGENET1K_V1 | ConvNet | torchvision | 73 | repghostnet_058.in1k | ConvNet | timm |
| 24 | MNASNet1_0_Weights.IMAGENET1K_V1 | ConvNet | torchvision | 74 | repghostnet_080.in1k | ConvNet | timm |
| 25 | MNASNet1_3_Weights.IMAGENET1K_V1 | ConvNet | torchvision | 75 | repghostnet_100.in1k | ConvNet | timm |
| 26 | MobileNet_V2_Weights.IMAGENET1K_V1 | ConvNet | torchvision | 76 | efficientvit_b0.r224_in1k | Transformer | timm |
| 27 | MobileNet_V2_Weights.IMAGENET1K_V2 | ConvNet | torchvision | 77 | efficientvit_b1.r224_in1k | Transformer | timm |
| 28 | RegNet_X_1_6GF_Weights.IMAGENET1K_V1 | ConvNet | torchvision | 78 | efficientvit_b2.r224_in1k | Transformer | timm |
| 29 | RegNet_X_1_6GF_Weights.IMAGENET1K_V2 | ConvNet | torchvision | 79 | efficientvit_b3.r224_in1k | Transformer | timm |
| 30 | RegNet_X_3_2GF_Weights.IMAGENET1K_V1 | ConvNet | torchvision | 80 | efficientvit_m0.r224_in1k | Transformer | timm |
| 31 | RegNet_X_3_2GF_Weights.IMAGENET1K_V2 | ConvNet | torchvision | 81 | efficientvit_m1.r224_in1k | Transformer | timm |
| 32 | RegNet_X_400MF_Weights.IMAGENET1K_V1 | ConvNet | torchvision | 82 | efficientvit_m2.r224_in1k | Transformer | timm |
| 33 | RegNet_X_400MF_Weights.IMAGENET1K_V2 | ConvNet | torchvision | 83 | efficientvit_m3.r224_in1k | Transformer | timm |
| 34 | RegNet_X_800MF_Weights.IMAGENET1K_V1 | ConvNet | torchvision | 84 | efficientvit_m4.r224_in1k | Transformer | timm |
| 35 | RegNet_X_800MF_Weights.IMAGENET1K_V2 | ConvNet | torchvision | 85 | efficientvit_m5.r224_in1k | Transformer | timm |
| 36 | RegNet_Y_1_6GF_Weights.IMAGENET1K_V1 | ConvNet | torchvision | 86 | coatnet_nano_rw_224.sw_in1k | Hybrid (Conv + Attention) | timm |
| 37 | RegNet_Y_1_6GF_Weights.IMAGENET1K_V2 | ConvNet | torchvision | 87 | coatnext_nano_rw_224.sw_in1k | Hybrid (Conv + Attention) | timm |
| 38 | RegNet_Y_3_2GF_Weights.IMAGENET1K_V1 | ConvNet | torchvision | 88 | seresnext101_32x4d.gluon_in1k | ConvNet | timm |
| 39 | RegNet_Y_3_2GF_Weights.IMAGENET1K_V2 | ConvNet | torchvision | 89 | vit_tiny_r_s16_p8_224.augreg_in21k | Transformer | timm |
| 40 | RegNet_Y_400MF_Weights.IMAGENET1K_V1 | ConvNet | torchvision | 90 | vit_small_r26_s32_224.augreg_in21k | Transformer | timm |
| 41 | RegNet_Y_400MF_Weights.IMAGENET1K_V2 | ConvNet | torchvision | 91 | vit_tiny_r_s16_p8_224.augreg_in21k_ft_in1k | Transformer | timm |
| 42 | RegNet_Y_800MF_Weights.IMAGENET1K_V1 | ConvNet | torchvision | 92 | vit_small_r26_s32_224.augreg_in21k_ft_in1k | Transformer | timm |
| 43 | RegNet_Y_800MF_Weights.IMAGENET1K_V2 | ConvNet | torchvision | 93 | hrnet_w18_small.gluon_in1k | ConvNet | timm |
| 44 | ResNeXt50_32X4D_Weights.IMAGENET1K_V1 | ConvNet | torchvision | 94 | hrnet_w18_small_v2.gluon_in1k | ConvNet | timm |
| 45 | ResNeXt50_32X4D_Weights.IMAGENET1K_V2 | ConvNet | torchvision | 95 | vit_small_patch16_224.dino | Transformer | timm |
| 46 | ResNet101_Weights.IMAGENET1K_V1 | ConvNet | torchvision | 96 | vit_base_patch16_224.mae | Transformer | timm |
| 47 | ResNet101_Weights.IMAGENET1K_V2 | ConvNet | torchvision | 97 | maxvit_tiny_tf_224.in1k | Hybrid (Conv + Attention) | timm |
| 48 | ResNet50_Weights.IMAGENET1K_V1 | ConvNet | torchvision | 98 | maxvit_tiny_rw_224.sw_in1k | Hybrid (Conv + Attention) | timm |
| 49 | ResNet50_Weights.IMAGENET1K_V2 | ConvNet | torchvision | 99 | vit_base_patch32_224.sam_in1k | Transformer | timm |
| 50 | ResNet34_Weights.IMAGENET1K_V1 | ConvNet | torchvision | 100 | vit_base_patch32_clip_224.openai_ft_in1k | Transformer | timm |

In Table 3, we include the full list of pre-trained models that are used in our evaluations. We include as follows relevant references and corresponding model indices in Table 3 (note that some pre-trained models correspond to multiple references):

- ResNet (He et al., 2015): 1, 46, 47, 48, 49, 50, 88;

- EfficientNet/EfficientNetV2 (Tan & Le, 2019; 2021): 2, 17, 18, 19, 20, 21;

- GoogLeNet (Szegedy et al., 2015): 3;

- Swin Transformer/Swin Transformer V2 (Liu et al., 2021; 2022b): 4, 54;

- MobileNet V2/V3 (Sandler et al., 2018; Howard et al., 2019): 5, 6, 7, 26, 27;

- MNASNet (Tan et al., 2019): 8, 23, 24, 25;

- ShuffleNet V2 (Ma et al., 2018): 9, 51, 52, 53;

- AlexNet (Krizhevsky et al., 2012): 10;

- ConvNeXt (Liu et al., 2022c): 11, 12, 87;

- DenseNet (Huang et al., 2017): 13, 14, 15, 16;

- Inception V3 (Szegedy et al., 2016): 22;

- RegNet (Radosavovic et al., 2020): 28, 29, 30, 31, 32, 33, 34, 35, 36, 37, 38, 39, 40, 41, 42, 43;

- ResNeXt (Xie et al., 2017): 44, 45, 88;

- Vision Transformer (Dosovitskiy et al., 2020): 55, 56, 57, 89, 90, 91, 92, 95, 96, 99, 100;

- Wide ResNet (Zagoruyko & Komodakis, 2016): 58, 59;

- MobileOne (Vasu et al., 2023): 60, 61, 62, 63, 64;

- InceptionNeXt (Yu et al., 2023): 65, 66, 67;

- GhostNet/GhostNetV2 (Han et al., 2020; Tang et al., 2022): 68, 69, 70, 71;
- RepGhostNet (Chen et al., 2022): 72, 73, 74, 75;
- EfficientViT (MIT) (Cai et al., 2022): 76, 77, 78, 79;
- EfficientViT (MSRA) (Liu et al., 2023): 80, 81, 82, 83, 84, 85;
- CoAtNet (Dai et al., 2021): 86, 87;
- Squeeze-and-Excitation (Hu et al., 2018): 88;
- Bag-of-Tricks (He et al., 2019): 88;
- AugReg (Steiner et al., 2021): 89, 90, 91, 92;
- HRNet (Wang et al., 2020): 93, 94;
- DINO (Caron et al., 2021): 95;
- Masked Autoencoder (He et al., 2022): 96;
- MaxViT (Tu et al., 2022): 97, 98;
- Sharpness-aware minimizer for ViT (Chen et al., 2021): 99;
- Reproducible scaling laws (Cherti et al., 2023): 100;
- CLIP (Radford et al., 2021): 100.

# B FIGURES OF EMPIRICAL EVALUATIONS

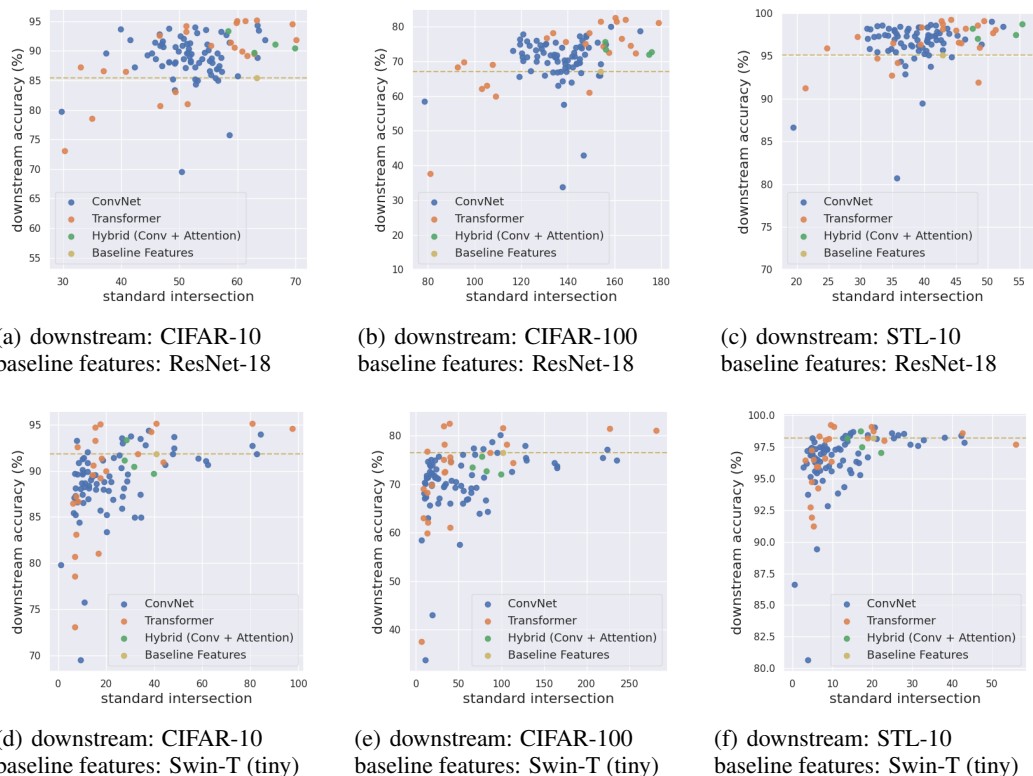

(a) downstream: CIFAR-10
baseline features: ResNet-18

(b) downstream: CIFAR-100
baseline features: ResNet-18

(c) downstream: STL-10
baseline features: ResNet-18

(d) downstream: CIFAR-10
baseline features: Swin-T (tiny)

(e) downstream: CIFAR-100
baseline features: Swin-T (tiny)

(f) downstream: STL-10
baseline features: Swin-T (tiny)

Figure 3: Downstream accuracy (i.e. the ground truth) v.s. the cardinality of standard intersections (i.e. the selection metric) when using 4k steps per candidate. The downstream accuracy of the baseline features are highlighted with the dashed line. When a public model is only suboptimal, using it as baseline features for Standardized Embedder can still locate more competitive models.

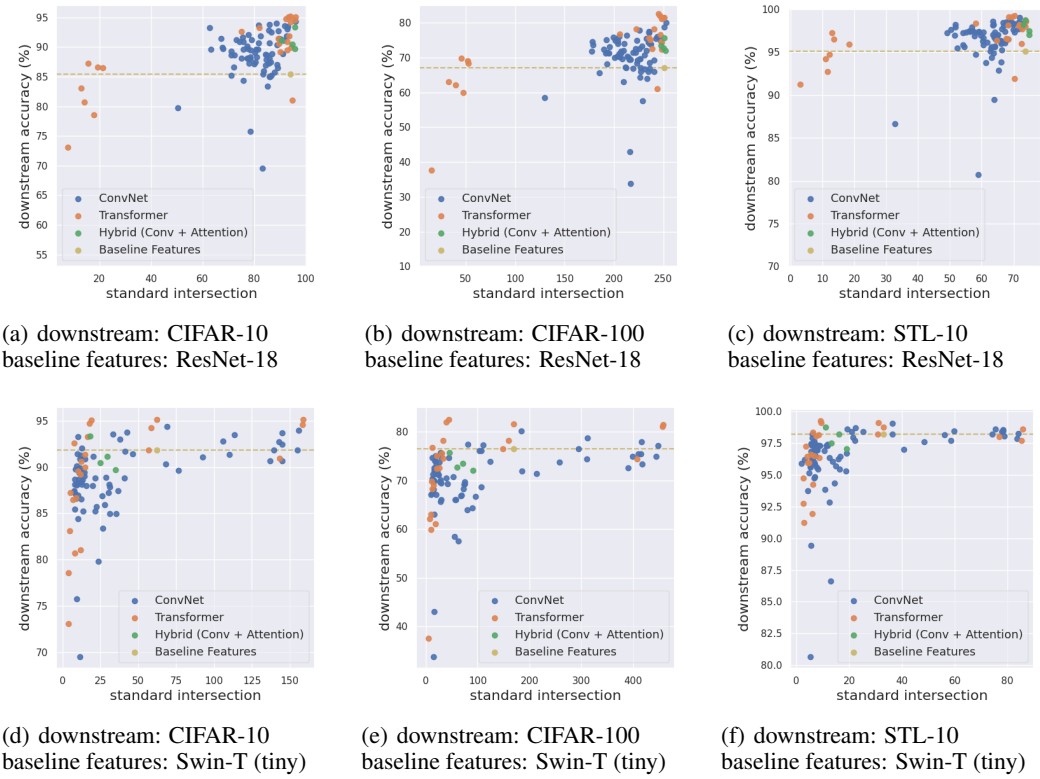

(a) downstream: CIFAR-10
baseline features: ResNet-18

(b) downstream: CIFAR-100
baseline features: ResNet-18

(c) downstream: STL-10
baseline features: ResNet-18

(d) downstream: CIFAR-10
baseline features: Swin-T (tiny)

(e) downstream: CIFAR-100
baseline features: Swin-T (tiny)

(f) downstream: STL-10
baseline features: Swin-T (tiny)

Figure 4: Downstream accuracy (i.e. the ground truth) v.s. the cardinality of standard intersections (i.e. the selection metric) when using 10k steps per candidate. The downstream accuracy of the baseline features are highlighted with the dashed line. When a public model is only suboptimal, using it as baseline features for Standardized Embedder can still locate more competitive models.

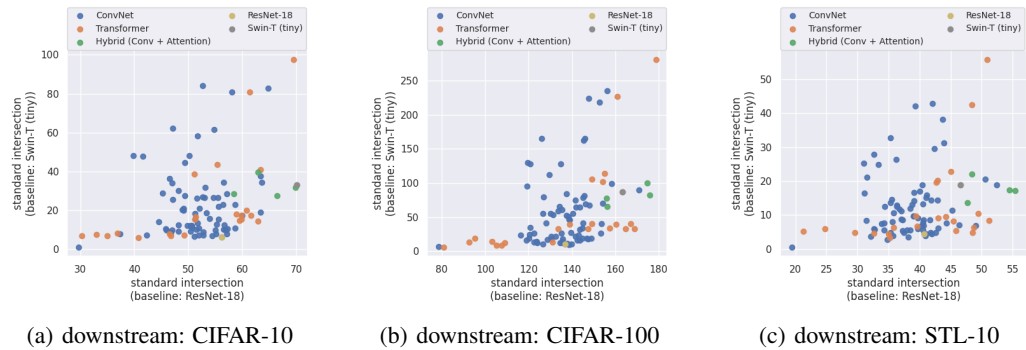

(a) downstream: CIFAR-10     (b) downstream: CIFAR-100     (c) downstream: STL-10

Figure 5: Comparing the cardinality of standard intersections (i.e. the selection metric) when using different baseline features (ResNet-18 and Swin-T (tiny)) with 4k steps per candidate. The green/orange points in the bottom right suggest using ResNet-18 as baseline features tend to over-estimate (some) models with attentions compared to using Swin Transformer (tiny).

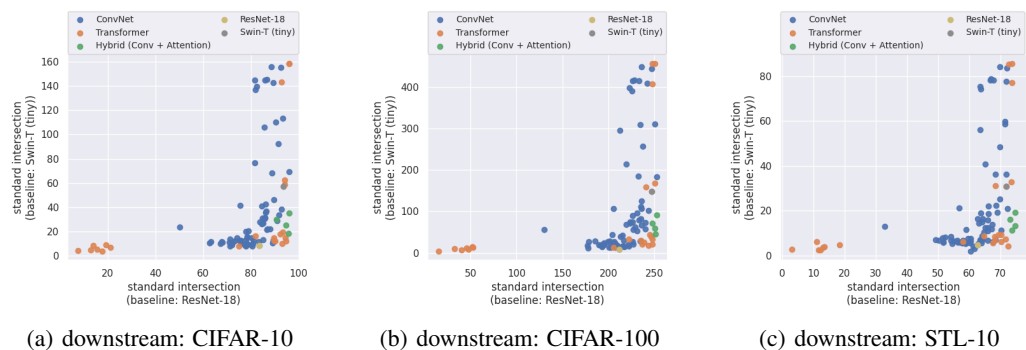

(a) downstream: CIFAR-10     (b) downstream: CIFAR-100     (c) downstream: STL-10

Figure 6: Comparing the cardinality of standard intersections (i.e. the selection metric) when using different baseline features (ResNet-18 and Swin-T (tiny)) with 10k steps per candidate. The green/orange points in the bottom right suggest using ResNet-18 as baseline features tend to over-estimate (some) models with attentions compared to using Swin Transformer (tiny).

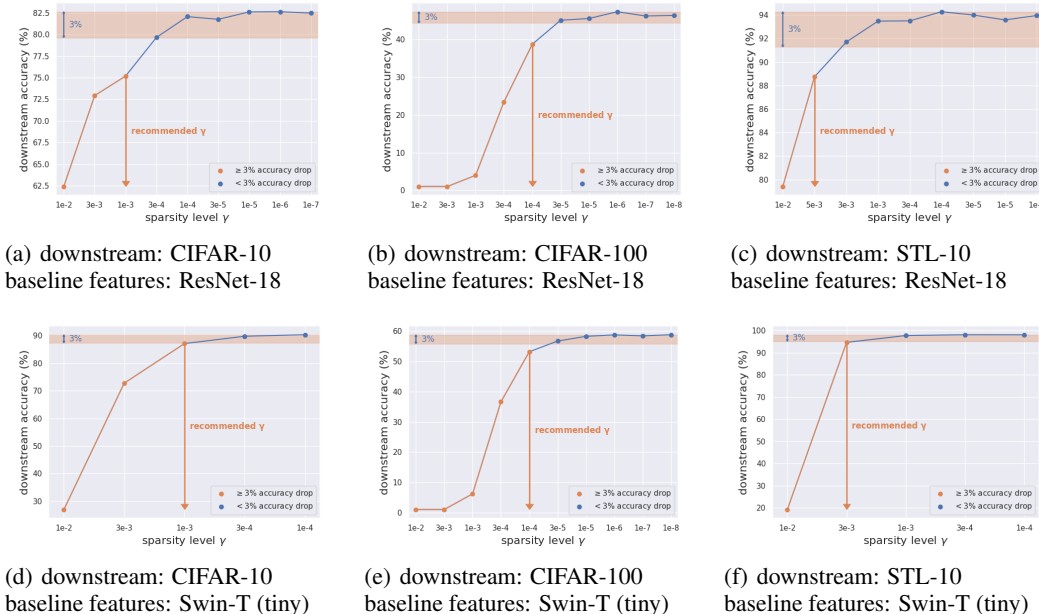

(a) downstream: CIFAR-10
baseline features: ResNet-18

(b) downstream: CIFAR-100
baseline features: ResNet-18

(c) downstream: STL-10
baseline features: ResNet-18

(d) downstream: CIFAR-10
baseline features: Swin-T (tiny)

(e) downstream: CIFAR-100
baseline features: Swin-T (tiny)

(f) downstream: STL-10
baseline features: Swin-T (tiny)

Figure 7: Downstream accuracy of the baseline features corresponding to varying level of sparsity regularization $\gamma$. A rule of thumb for deciding the value of $\gamma$: using the smallest $\gamma$ with at least 3% accuracy drop from the converged accuracy.

## C    ILLUSTRATIVE EXAMPLE FOR SECTION 4.2

Here is an example to illustrate how to associate every subset of the baseline feature set $B$ with a binary vector from $\{0, 1\}^N$. Assuming now the baseline feature set $B : \mathcal{X} \to \mathbb{R}^N$ contains a total of $N = 4$ features, $b_0, b_1, b_2, b_3$, where each of them is a function from $\mathcal{X}$ to $\mathbb{R}$, then there will be a total of $2^4 = 16$ different subsets of $B$. We can associate each subset with a distinct, 4-dimensional binary vector (i.e. a vector in $\{0, 1\}^4$) by using $1$ to indicate the presence of a feature and $0$ to indicate an absence of a feature in the subset. Specifically, $(0, 0, 0, 0)$ will denote the empty subset, $(0, 0, 1, 0)$ will denote $\{b_2\}$ and $(1, 0, 1, 1)$ will denote $\{b_0, b_2, b_3\}$.

## D    LIMITATIONS

While the proposed paradigm, independently-prepared query-efficient model selection, and the proof-of-concept solution, Standardized Embedder, are both very general, there are some limitations of our experiments worth noting. Each of them corresponds to possible future directions.

- Limitation of input modality: In our experiments, we evaluate on vision models with image inputs. We expect this can be generalized to other modalities including but not limited to text, audio, tabular data and time series in the future.

- Limitation of types of downstream tasks: In our experiments, we evaluate on downstream classification tasks. We expect this can be generalized to other type of tasks including but not limited to segmentation, detection, clustering and reconstruction in the future.

- Limitation of linear probing: In our experiments, we use linear probing to approximate ground truth accuracy. We expect other transfer learning approaches to be incorporated in the future, including but not limited to full fine-tuning, partial fine-tuning and ProtoNet (Snell et al., 2017).

