# OpenReview forum: "Independently-prepared Query-efficient Model Selection"
_ICLR.cc/2024/Conference — ICLR 2024 Conference Withdrawn Submission_

### Official Review · Reviewer_EJZF · 2023-10-23

**Soundness:** 2 fair
**Presentation:** 2 fair
**Contribution:** 2 fair
**Rating:** 5
**Confidence:** 5

**Summary:**

This paper focuses on the problem of model selection, i.e., selecting one or few models from a large pre-trained model pool for downstrem task fine-tuning with best performance. The paper advocates for a new model selection paradigm with two merits: query-efficient and independently-prepared. This paper also proposes a model selection method named Standardized Embedder under this paradigm. The experimental results demonstrate its effectiveness.

**Strengths:**

- The proposed model selection paradigm (independently-prepared & query-efficient) is good.
- The proposed Standardized Embedder is effective. The idea of standardization is interesting.
- The related works are clearly introduced.

**Weaknesses:**

- The description of the proposed method is hard to understand. I read section 4 for multiple times and tried to figure out the implementation details of Standardized Embedder, but I still have some questions.
  - I don't understand "any subset of the baseline feature set B can be directly associated with a binary vector $\{0,1\}^N$". Definition 2 is thereby unclear to me.
  - I think there may be a typo related to $\delta$-equivalence. In **Definition 1**, $\delta$-equivalence means the expected similarity between transformed features is larger than $1-\delta$, therefore if the $\delta$ is small, then the similarity is high (near to one). However, in the last line in page 6, "the optimization wants to find a subset of the baseline features that is $\delta$-equivalent to $F$ for larger $\delta$". I think the "larger $\delta$" should be corrected as smaller $\delta$.
  - Although I understand the idea of standardization (using baseline features to derive the features of models and tasks), there are still other details of this method, and why are them designed in this way? For example,
    - Why is $v$ constrained in $[0,1]^n$?
    - Why is the equivalence defined using affine transformations? For example, why don't we use $w^TF(x)$ only?
- The design of Standardized Embedder lacks enough justification. For example,
  - Are there any theoretical justification for the standardization? Although it's intuitive, the theoretical analysis is necessary to explain the effectiveness.
  - The comparison between Standardized Embedder and other existing model selection methods are missing.
  - The study on task distribution generalization is missing. It's unclear if the selection works for task distributional shifts in downstream tasks. For example, as the current datasets are all about classification - what about the selection performance on other tasks like segmentation or object detection?
- There are some weaknesses about the experimental settings.
  - The "underlying distribution D" is approximated using ImageNet, which has class overlap with CIFAR (downstream task dataset).
  - The models are trained with a linear classifier and fixed feature. This is usually different with downstream fine-tuning. The results may thereby not be applicable to real applications as full model fine-tuning usually results in better results.

**Questions:**

- How to define "feature" for different models? As a neural networks usually contain many layers and different layer outputs may be used to construct the mapping $F: \mathcal X\to \R^n$, it is unclear how to choose it for different models. I guess in this paper, the final layer outputs before classification head are treated as feature. But what about the models like GPT-2 or BART that have many potential "features" definition (special token embeddings, avg/max-pooling of token embeddings, decoder/encoder representations)?

- In Definition 1, is $\delta\in [0,2]$?

---

> ### Author Response · Authors · 2023-11-14
>
> Thank you for reviewing our paper!
>
> We are glad to know that you like both the new model selection paradigm and the idea of standardization. We also apologize as you find some description hard to understand: We are here to address any questions from you and will use them to improve the clarity and quality of our paper during revisions.
>
> **1. I don't understand "any subset of the baseline feature set B can be directly associated with a binary vector $\\{0,1\\}^N$."**
>
> Please allow us to explain this with a toy example. Assuming now the baseline feature set contains a total of $N=4$ features, $b_0, b_1, b_2, b_3$, then there will be a total of $2^4=16$ different subsets of B, and we can associate each subset with a distinct, 4-dimensional binary vector (i.e. a vector in $\\{0,1\\}^4$). Specifically, (0, 0, 0, 0) will denote the empty subset, (0, 0, 1, 0) will denote $\\{b_2\\}$ and (1, 0, 1, 1) will denote $\\{b_0, b_2, b_3\\}$, i.e. each 1 indicating the presence of a feature and each 0 indicating an absence.
>
> **2. I think there may be a typo related to $\delta$-equivalence.**
>
> Good catch! This is indeed a typo and we will have it fixed: The optimization maximizes cosine similarity and therefore it is trying to find a subset of the baseline features that is $\delta$-equivalent to F for **smaller** $\delta$.
>
>
> **3. Why is $v$ constrainsted in $[0, 1]^n$?**
>
> Because $v$ is a continuous relaxation of a binary vector from $\\{0, 1\\}^n$. As we clarified in part 1 of our response for you, each subset of the baseline features can be associated with a binary vector from $\\{0, 1\\}^n$ (i.e. every dimension is either 0 or 1). The idea of continuous relaxation is to use $v$, a vector with each dimension constrained to be any value between 0 and 1, as a surrogate for binary vectors defined in Definition 2 so that the overall optimization problem will be continuous instead of discrete.
>
> **4. Why is the equivalence defined using affine transformations? For example, why don't we use $w^\top F(x)$ only?**
>
> That is a good question! Conceptually, how we should define equivalence of features depends on our knowledge/assumption regarding how these features will be used. Using affine transformations is because **in most use cases of pre-trained features, there will be at least one linear layer and at least one linear layer will have a learnable bias term**. For example, say now there will be one linear layer $g_{w_g, b_g}(x) = w_g^\top x + b_g$ attached when using features (there may be more layers after g), then we have $g_{w_g, b_g}(w^\top F(x) + b) = g_{w w_g, b_g + w_g^\top b}{F(x)}$, meaning that any functions representable by a linear layer (with bias) after $w^\top F(x) + b$ will also be representable by a linear layer (with bias) after $F(x)$. We will include explanations like this in our paper to make it easier to understand.
>
> **5. Are there any theoretical justification for the standardization? Although it's intuitive, the theoretical analysis is necessary to explain the effectiveness.**
>
> We consider our proposed method, standardized embedder, as an theoretically motivated approach, meaning that it is derived through theoretical tools and theoretical analysis but there is no rigorous theoretical guarantee of effectiveness. Respectfully, we argue ‘a rigorous theoretical guarantee of effectiveness’ is valuable but not necessary, given that no existing model selection work (as we discussed in Section 2 Related Work) offers a guarantee of such.
>
> **6. The comparison between Standardized Embedder and other existing model selection methods are missing**
>
> The main subject of this paper is NOT standardized embedder BUT the new paradigm, i.e. independently-prepared query-efficient model selection. As we emphasize in our paper, standardized embedder is a proof-of-concept solution to showcase that the new paradigm is possible.
>
> The primary benefit of the new paradigm (compared to existing ones) is that it has fundamentally better preparation and query complexity & scalability with respect to increasing numbers of candidate models. These originate from the paradigm design and are elaborated in Section 3 of our paper.
>
> Consequently, the purpose of our proof-of-concept solution and our experiments is to show that such a paradigm is realizable and feasible. Since existing model selection methods do not fit in the new paradigm, we argue that the existing experiments should be sufficient for this purpose.

---

> > ### Author Response · Authors · 2023-11-14
> >
> > **7. The study on task distribution generalization is missing. It's unclear if the selection works for task distributional shifts in downstream tasks.**
> >
> > Regarding task generalization, we agree that including other tasks can be interesting. However, given that many previous works also focus on the scope of classification tasks, we believe a demo in image classifications should be acceptable as supports to the feasibility of the new paradigm.
> >
> > **8. The "underlying distribution D" is approximated using ImageNet, which has class overlap with CIFAR (downstream task dataset).**
> >
> > We understand your concern but we argue that this should not be considered a critical issue. This is very similar to the pretraining dataset and downstream dataset in self-supervised pre-training of vision models (e.g. MAE, SimCLR, MoCo) if the reviewer is familiar with them: Obviously we do not expect our method to work if the distribution D we used in preparations is completely irrelevant to the downstream distributions, but as long as we can use a fairly broad distribution as D and generalize to its relevant domains, the method can be useful.
> >
> > Thus we argue that even though how to have a better characterization on the degree of allowed distribution shifts remains open, the provided evaluations should be sufficient, at least for the purpose of supporting the potential of the new paradigm.
> >
> > **9. The models are trained with a linear classifier and fixed feature. This is usually different with downstream fine-tuning. The results may thereby not be applicable to real applications as full model fine-tuning usually results in better results.**
> >
> > We agree that other fine-tuning methods can result in different accuracy from linear probing. Here we use linear probing as a **representative** of downstream performance due to practical constraints:
> > When using other fine-tuning methods, the downstream performance of pre-trained models are potentially very sensitive to the hyper-parameters and the choice of training recipes. Given the diversity of the pre-trained models evaluated, it is unfortunately computationally too heavy for us to iterate through different training recipes and conduct hyper-parameter search for every pre-trained model individually (which is an example of the inefficiency of naive mode selection). Meanwhile, linear probing offers a representative downstream accuracy that is fairly insensitive to hyper-parameters.
> >
> > **10. How to define "feature" for different models?**
> >
> > Good question! While in our experiments we use the penultimate layer as features, in principle, one can use any layers of a model, depending on how the model will be used in downstream tasks. For example, if there is a pre-trained ResNet-50, one can have multiple definitions of ‘feature’: If I define the penultimate layer as features, then I should use the penultimate layer as features for downstream tasks; If I define a layer in the middle (with the last block of ResNet-50 removed) as features, I should also use that layer as features in downstream tasks.
> >
> > **11. In Definition 1, is $\delta\in[0, 2]$?**
> >
> > In some sense, yes. Since cosine similarity is bounded in $[-1, 1]$, no pair of feature sets can be $\delta$-equivalent for $\delta<0$ and any pair of feature sets will be $\delta$-equivalent for $\delta>2$.
> >
> > **Please let us know your comments at your earliest convenience.
> > Thanks again for reviewing!**

---

> > ### Comment · Reviewer_EJZF · 2023-11-15
> >
> > Dear Authors,
> >
> > Thank you for your detailed explanations. Despite your responses, I still have several concerns:
> >
> > 1. Your explanation is appreciated. Could you revise the manuscript to make this section clearer and more interpretable?
> >
> > 2. You can fix the typo of delta-equivalence in your manuscript, as ICLR allows for paper revision.
> >
> > 3&4. These points are satisfactory.
> >
> > 5. Since I have many concerns in the experimental settings, the lack of theoretical justification for the standardization is a weakness of this paper for me.
> >
> > 6. While your explanation is helpful, proposing a new paradigm without a robust technique limits its impact. The Standardized Embedder, though a step forward, doesn't convincingly demonstrate significant benefits. I suggest adding a comparative analysis with traditional model selection methods to underline the importance of your new paradigm.
> >
> > 7. Minor point, but additional experiments on task distributional generalization could significantly bolster the case for the Standardized Embedder.
> >
> > 8. No further comments.
> >
> > 9. The sensitivity of fine-tuning to hyperparameters and resource demands are understood. However, for real-world applicability, it's essential to demonstrate the method's effectiveness with full fine-tuning, not just linear probing. If running comprehensive experiments is challenging, consider using a smaller model subset.
> >
> > 10. As we can define feature using different layer outputs, how to select the feature according to your method?
> >
> > 11. No further comments on this point.
> >
> > In conclusion, while the new paradigm introduced in your paper is meaningful, the current version doesn't sufficiently demonstrate the effectiveness of the proposed method.

---

> ### Author Response · Authors · 2023-11-16
> **Thank you for responding to our rebuttal!**
>
> Thank you for responding to our rebuttal!
>
> ---
>
> **(1)** For 1&2, we just updated the manuscript. Note that for 1, we add the illustrative example to Appendix C due to space limits and refer to it in Section 4.2.
>
> ---
>
> **(2)** For 3&4&8&11, thank you for letting us know that these are addressed!
>
> ---
>
> **(3)** For 5&6&7&9, we wonder if the reviewer is currently considering only a narrow definition of impact that excludes our primary contributions. To be specific, we admit that the current evaluations only support the effectiveness of standardized embedder in the context of image classification and linear probing, however, we argue that it is more than sufficient as a proof-of-concept to support the feasibility of the new paradigm and **more importantly, a new paradigm without a robust technique does NOT necessarily limit its impact.**
>
> Here is **an example to illustrate the structure of scientific revolutions**: It is well-known that the Wright brothers invented the first airplane in 1903, which is certainly a paradigm changer since it demonstrates the possibility of heavier-than-air, sustained, powered and controlled aircraft. However, this plane is not really a robust technique, given that it survived only 4 brief, low-altitude flights with a maximum distance of 852 ft (260 m) before it got damaged in landing and minutes later blew over by a heavy gust. It was not until 36 years later, in 1939, that we had the first jet aircraft, and not until another 13 years later, in 1952, that we had the first jet airliner.
>
> Obviously, we are not claiming that our new model selection paradigm is as important as the first airplane or the future progress based on our paradigm will be as difficult as inventing the first jet aircraft. We just hope more diverse definitions of contributions can be considered. In the context of this submission, we argue that the impact of the submission is already sufficient since the new paradigm offers benefits that are critical to applications and the proof-of-concept solution supports the feasibility/realizability of the new paradigm.
>
> ---
>
> **(4)** In addition, for 5, if possible, could you please provide an example reference of model selection with the degree of theoretical justifications that you find sufficient? We believe provable model selection can itself be a very promising research project and will be very interested in any relevant work.
>
> ---
>
> **(5)** For 10: The short answer is simply that one can consider different feature layers of the same pre-trained model as different models. If you recall, we define each model as a set of features when introducing our method in Section 4. For example, if I have a pretrained model containing two layers $f_1: \mathcal{X} \to \mathbb{R}^{d_1}$ and $f_2:\mathbb{R}^{d_1} \to \mathbb{R}^{d_2}$, then I simply can have two different feature sets depending on which layer will be used as features: One is the composition of all layers $f_2\circ f_1: \mathcal{X} \to \mathbb{R}^{d_2}$, and another one simply contains a single layer $f_1: \mathcal{X} \to \mathbb{R}^{d_1}$. They can be treated as two separate models when using standardized embedder.
>
> ---
>
> Thanks again! Please let us know if anything. Your prompt feedbacks will always be appreciated!

---

> > ### Comment · Reviewer_EJZF · 2023-11-19
> >
> > Dear Authors,
> >
> > Thank you for your feedback. I'll start with your most recent comment.
> >
> > Regarding point (5): Apologies for any confusion. I'm interested in understanding how to select the most effective feature layer from a pre-trained model for optimal performance. Could you provide more insights on this?
> >
> > For point (4): My query is not about the theoretical underpinnings of **model selection**, but rather about the specific justifications for your proposed "Standardized Embedder" method.
> >
> > Concerning point (3): I appreciate your detailed discussion. I am not excluding your primary contributions.
> >
> > Having reviewed your paper and rebuttal thoroughly, I understand that your main contributions are twofold: introducing a new model selection paradigm and the "Standardized Embedder". My focus on the latter does not negate the importance of the former. However, I still have unresolved queries regarding the "Standardized Embedder".
> >
> > As you emphasize the new paradigm as a key contribution, let's delve deeper into it, setting aside the specific technique for now. Here are my comments:
> >
> > 1. How vital is privacy preservation in your paradigm? Once a model is selected through your algorithm, full access for training is necessary. How do you address this in practical applications?
> > 2. Converting a model into embeddings might entail loss of significant model information due to compression.
> > 3. The paradigm appears to merge two existing ones, drawing heavily on the concept of model embedding.
> > 4. In the description of 'Flexibility', what does it mean by "...each of Figure 1(a), 1(b), and 1(c) contains 1510 × 1170 ≈ 1.8 × 10^6 RGB pixels, or about 5.3 × 10^6 dimensions"?
> > 5. While I agree that model selection is critical, it seems your proposed paradigm still operates within the traditional framework of preparations and queries.
> > 6. It seems that the limitations of this paradigm and the proposed methods are not discussed in the paper.
> >
> > I look forward to your responses to these points.

---

> ### Author Response · Authors · 2023-11-20
>
> Dear reviewer,
>
>
> We sincerely appreciate your active participation in the discussion!
>
>
> We will cover the points following the order they are mentioned in your latest reply.
>
>
> ---
> **For point (5): ‘I'm interested in understanding how to select the most effective feature layer from a pre-trained model for optimal performance.’**
>
>
> We rephrase the question a little bit: Given a task and a pre-trained model, how can one select a layer of the pre-trained model that is the most useful for this task? Obviously, the answer depends on both the design of the model and the task. A common practice nowadays is to use the penultimate layer, because it is found to be working fine in many cases (which is why we follow this in our experiments). However, obviously this is not necessarily the optimal choice and **in many cases so far, people have to select a layer based on some heuristics**: For example, if it is known that the task depends heavily on spatial information (e.g. similar objects/features in different locations may mean different things), one should probably use features before pooling instead of the penultimate layer (where location information is removed by global pooling) for a ResNet model.
>
>
> One can still select layers based on heuristics for model selection, but if one prefers automatic solutions, our method may provide a solution (by embedding multiple layers and selecting as if they are different models). This can be another potential benefit of Standardized Embedder (even though we do not emphasize it in our paper as it is relatively minor).
>
>
>
>
> ---
> **For point (4): My query is not about the theoretical underpinnings of model selection, but rather about the specific justifications for your proposed "Standardized Embedder" method.**
>
>
> Could you please advise on why you find the theoretical justification of Standardized Embedder insufficient? We are a bit confused on this front, because we believe we offer a similar/higher level of theoretical justifications as existing work. For example, Task2Vec/Model2Vec [1] provides a fairly similar level of theoretical justifications as ours in a sense that their method is also derived with theoretical motivations but with no theoretical guarantee of success. We believe Task2Vec/Model2Vec [1] is typically considered a good work in this field.
>
>
>
>
> reference:
> [1] Achille, Alessandro, et al. "Task2vec: Task embedding for meta-learning." Proceedings of the IEEE/CVF international conference on computer vision. 2019.
>
> ---
> **For point (3):**
>
>
> Thank you for the detailed response and your new comments!
> We will start addressing them for you now.
>
> **1.How vital is privacy preservation in your paradigm? Once a model is selected through your algorithm, full access for training is necessary. How do you address this in practical applications?**
>
> To clarify, the proposed paradigm do not remove the requirement of (eventually) accessing the **selected model**. As detailed in Section 3.2, the privacy benefits are twofold: (1) Model owners only share embeddings and their models are kept private **during selection**. (2) The users (who want to select models for their tasks) can keep their data to their own and they do not need to inform model owners about the selection. **After a model is selected**, the user contacts **only** the owner of the selected model to gain access (for example, to purchase the model).
>
> Comparing to existing paradigms, an important value of ours is that model owners do not need to share their models to any others just to get considered in model selections: In the case of distributed model markets, this means that model owners only grant access to actual buyers (that have selected their models) and potential buyers do not need to purchase models they do not want just for model selection.
>
> **2.Converting a model into embeddings might entail loss of significant model information due to compression.**
>
> Indeed, this is exactly why we build a proof-of-concept solution to show that it is possible to keep sufficient information to select models that perform well. Obviously, we are not saying that every method following the new paradigm should be effective. What we argue is that there are methods that can be effective while enjoying the benefits of the new paradigm and therefore the new paradigm is promising.

---

> > ### Author Response · Authors · 2023-11-20
> >
> > **3.The paradigm appears to merge two existing ones, drawing heavily on the concept of model embedding.**
> >
> > We understand why such confusion may appear but we respectfully disagree. As illustrated in Figure 1, the two existing paradigms suffer from scalability issues for different stages: One of them suffers from inefficient queries and the other one suffers from poorly-scalable preparations. The high-level goal of the new paradigm is to avoid/mitigate issues in both stages but that is very different from ‘merging two existing ones’.
> >
> > We agree that the new paradigm can be seen as a small modification to the paradigm of query-efficient model selection (by introducing independent preparations). However, we argue that this small change matters a lot in a sense that (1) This small change brings us many benefits as we presented in Section 3.2; and (2) Existing solutions cannot be easily adapted to satisfy this small change. Thus we argue this should not be considered a weakness.
> >
> > **4.In the description of 'Flexibility', what does it mean by "...each of Figure 1(a), 1(b), and 1(c) contains 1510 × 1170 ≈ 1.8 × 10^6 RGB pixels, or about 5.3 × 10^6 dimensions"?**
> >
> > This is a comparison to help readers understand qualitatively how portable model embeddings typically are. In our experiments, the model embedding vectors are less than 1000 dimensions. A vector of this size can be easily transferred and/or broadcasted through the modern internet. We illustrate this by comparing it with (the dimensions of) the figures in our own submission.
> >
> > **5. While I agree that model selection is critical, it seems your proposed paradigm still operates within the traditional framework of preparations and queries.**
> >
> > We believe there might be some confusion here. Preparations are basically pre-processing and they can be used to denote any computations that can be done without downstream-specific/task-specific information. Queries are essentially any steps that one does after the downstream data are given to select models. Thus it is totally reasonable that our new paradigm still consists of preparations and queries.
> >
> > **6. It seems that the limitations of this paradigm and the proposed methods are not discussed in the paper.**
> >
> > While we believe we did not overclaim the implication of the proposed method in our submission, we agree discussions regarding the limitation of our experiments can be valuable. We have updated our manuscripts to include such discussions. Due to space limits and time limits (of the rebuttal), currently we add a discussion of limitations in Appendix and refer to it in Conclusion. We will try to fit it into the main text later when time permits. Thanks for your suggestion and understanding!
> >
> > ---
> >
> > Thank you for discussing with us and helping us to improve our paper.
> >
> > We also look forward to hearing back from you! Your comments are always welcome.

---

> > > ### Comment · Reviewer_EJZF · 2023-11-20
> > >
> > > Dear Authors,
> > >
> > > Thank you for your prompt feedback. Your responses addressed many of my concerns, and I would like to raise my score to 5.

---

> > > > ### Author Response · Authors · 2023-11-20
> > > >
> > > > Dear reviewer,
> > > >
> > > > We really appreciate your prompt feedbacks and the amount of time you must have spent on reviewing our paper.
> > > > However, we are very confused about your adjusted score for our submission.
> > > >
> > > > While we understand the judgement regarding potential impacts of a new paradigm can usually be challenging and we respect your opinion, could you please let us know what the primary remaining concerns are so that you consider our work 'below the acceptance threshold'?
> > > >
> > > > Thank you again for your participation in discussion! Looking forward to your reply.

---

> > > > > ### Comment · Reviewer_EJZF · 2023-11-20
> > > > >
> > > > > Dear Authors,
> > > > >
> > > > > Thank you for your submission. After careful consideration, these key points contribute to my assessment that this work falls below the acceptance threshold:
> > > > >
> > > > > 1. The lack of validation of the proposed technique in fine-tuning scenarios raises concerns about its applicability in real-world settings.
> > > > > 2. Consequently, it's premature to describe the technique as a proof-of-concept solution for real-world model selection, given the current limitations in its applicability.
> > > > > 3. There are other unaddressed points in my responses. For example, an empirical comparison with other existing model selection baselines in a real applications, and the generalization to different tasks. I also notice that the authors cannot give a time consumption analysis in the response to Reviewer SGa4. Although conducting comprehensive experiments may be challenging, such depth is crucial for substantiating the work's validity.
> > > > >
> > > > > Please be assured that my final evaluation is not arbitrary but the result of an in-depth and comprehensive review, reflecting my understanding of the community's standards and the expected level of contribution for such work. I hope you find this feedback understandable and informative.

---

> ### Author Response · Authors · 2023-11-20
>
> Thank you so much for getting back to us.
>
> Please allow us to provide additional information and clarifications for the concerns you just mentioned.
>
> ---
>
> 1. We agree with you that adding evaluations with fine-tuning can strengthen the experiments of this work. However, we want to add that fine-tuning is not necessarily the go-to option in practice and linear probing is still useful. For example, it is suggested that linear probing usually offers better out-of-distribution generalizations than fine-tuning, per the evaluation by Kumar et al.
>
> reference: Kumar, Ananya, et al. "Fine-Tuning can Distort Pretrained Features and Underperform Out-of-Distribution." International Conference on Learning Representations. 2022.
>
> ---
>
> 2. To clarify, we do not describe the technique as a proof-of-concept solution for real-world model selection. We describe the technique as a proof-of-concept solution for our paradigm, or more specifically, it is a proof-of-concept for the following argument: One can learn model embeddings that are indicative of downstream performance while keeping the embedding process independent for different models.
>
> ---
>
> 3. To clarify, here is how we claim the advantage of our paradigm (and therefore our proof-of-concept solution) compared to existing ones: With a sufficiently large number of candidate models (say M candidate models), our paradigm (and our solution), is the only one so far that can scale. While transferability metrics require O(M) model operations per query and Model2Vec requires updating O(M) existing model embeddings when a new candidate joins, Standardized embedder requires a constant amount of (i.e. O(1)) model operations per query and update only 1 model embedding when a new candidate joins (which is the embedding corresponding to the new candidate). This is also why we believe the computation analysis we shared with Reviewer SGa4 should be sufficient to demonstrate our advantage in scalability.
>
> ---
>
> Lastly, while obviously we as authors believe in our own work, we respect the reviewer's judgment and hope these new clarifications can be helpful towards a well-informed decision.
>
> Thank you for your services!

---

### Official Review · Reviewer_BVf9 · 2023-10-30

**Soundness:** 4 excellent
**Presentation:** 4 excellent
**Contribution:** 4 excellent
**Rating:** 8
**Confidence:** 5

**Summary:**

With the increasing number of open-sourced models in the community, model selection becomes more and more important. However, the scalability of existing solutions is limited with the increasing amounts of candidates. This work presents a new paradigm for model selection, namely independently-prepared query-efficient model selection. The advantage of their paradigm is twofold: first, it is query-efficient, meaning that it requires only a constant amount of model operations every time it selects models for a new task; second, it is independently-prepared, meaning that any information about a candidate model that is necessary for the selection can be prepared independently requiring no interaction with others. Consequently, the new paradigm offers many desirable properties for applications: updatability, decentralizability, flexibility, and certain preservation of both candidate privacy and query privacy. With the benefits uncovered, they present Standardized Embedder as a proof-of-concept solution to support the practicality of the proposed paradigm. Empirical results across different model architectures and various training recipes highlight the potential of the proposed paradigm.

**Strengths:**

- Model selection is an interesting, timely and important topic with the increasing number of open-sourced models in platforms such as HuggingFace.
- The proposed paradigm is novel and technically sound with various desirable properties, such as updatability, decentralizability, flexibility, and certain preservation of both candidate privacy and query privacy.
- The proposed method effectively selects a suitable model from a pool of 100 models and achieves good performance on downstream tasks.
- The paper is well-written and easy to follow.

**Weaknesses:**

- The existing work [1] seems to evaluate the pool of 1000s models. The scale of the number of models could be further increased.

[1] Bolya, Daniel, Rohit Mittapalli, and Judy Hoffman. "Scalable diverse model selection for accessible transfer learning." *Advances in Neural Information Processing Systems* 34 (2021): 19301-19312.

- A typo of strike-out text in Section 5.2.

**Questions:**

- What is the impact of using different models for baseline features?
- What are other potential applications of this new paradigm of model selection?

---

> ### Author Response · Authors · 2023-11-14
>
> Thank you so much for liking our paper!
> It is really a delight to have our contributions recognized!
>
> Meanwhile, we will be doing our best in providing responses to your questions.
>
> **1. The existing work [1] seems to evaluate the pool of 1000s models. The scale of the number of models could be further increased.**
>
> [1] is already mentioned in our paper. Notably, they evaluate a pool of 65 models instead of 1000s models, despite the fact that they mention ‘100s and 1000s of source models’ when introducing their motivations. That being said, we still agree with you that further increasing the number of models can further strengthen our empirical results, at least to some extent. However, we also want to recap that the most important contribution of this submission is the new model selection paradigm rather than the proof-of-concept solution to support its feasibility. We believe the scale of current evaluations is sufficient in this context.
>
>
> reference: [1] Bolya, Daniel, Rohit Mittapalli, and Judy Hoffman. Scalable diverse model selection for accessible transfer learning
>
> **2. A typo of strike-out text in Section 5.2.**
>
> Great catch! Thank you for reading our paper carefully!
>
> **3. What is the impact of using different models for baseline features?**
>
> This is a good question. We use Section 5.3 of our paper to discuss exactly the topic. In summary, from what we observe in Figure 5 and Figure 6, it is likely that models with attention, such as Swin Transformer (tiny) we evaluated, can be better choices of baseline features, because the reception fields of CNN may pose some limitations in characterizing some long-range correlations. However, it remains open regarding exactly what kind of models will serve best as baseline features (i.e. **we know there are likely many good choices, including the ones evaluated, but we do not know yet which should be the best**).
>
> This is potentially one future direction for us.
>
> **4. What are other potential applications of this new paradigm of model selection?**
>
> The essence of the new paradigm is to enable independent embedding of different models, which makes the entire selection scheme updatable and decentralizable. Building on these, there may be potential applications like: (1) vectorized database for models; (2) web search engines that search model embeddings; (3) decentralized model markets (e.g. on blockchain).
> Of course this is unlikely to be an exhaustive list.
>
> **Thank you again for recognizing our contributions!
> Please let us know if the response helps. Any follow-up questions are welcomed!**

---

> > ### Comment · Reviewer_BVf9 · 2023-11-20
> > **Post-rebuttal response**
> >
> > The reviewer would like to thank the detailed responses from the authors. The authors present a very interesting perspective on model selection. My concerns have been addressed, and I will keep my original score and vote for the acceptance.

---

> > > ### Author Response · Authors · 2023-11-20
> > > **Thank you!**
> > >
> > > Thank you, reviewer BVf9!
> > >
> > > It really means a lot to us that you find our work interesting!
> > >
> > > Have a good day!

---

### Official Review · Reviewer_SGa4 · 2023-11-03

**Soundness:** 1 poor
**Presentation:** 2 fair
**Contribution:** 1 poor
**Rating:** 3
**Confidence:** 5

**Summary:**

The authors' research belongs to a classical problem in the field of AutoML - automatic model selection. The two existing paradigms for model selection are Preparation-free and Query-efficient, but the existing paradigms do not have good scalability and efficiency in the face of the current massive requirements and models. So the authors propose a new paradigm named Independently-prepared Query-efficient, which makes the model recommendation process more accurate and efficient by computing embedded vectors for each model to be selected independently.

**Strengths:**

1. The authors present a very interesting perspective and way of thinking, starting from the paradigm of model selection, where each model to be selected is represented by an embedded vector and is computed and updated independently. In this way, the efficiency and accuracy of model recommendation can be improved, and the authors argued the decentralizability and flexibility of the new paradigm.
2.  The authors give a theoretical proof for the newly proposed STANDARDIZED EMBEDDER, which justifies from a theoretical point of view the deflation of the tensor representation of the depth space.
3. The authors conducted a series of experiments on a commonly used dataset for image categorization, and the experimental results demonstrated the realization of SOTA in terms of accuracy.

**Weaknesses:**

1.  I think the author has a problem with the title of the paper and the writing of the whole paper. The author says that he has proposed a new paradigm, but he has only made a little modification in the computation and mapping of embedded vectors, which does not match the innovation of the "new paradigm", and it is more suitable to use the "new method" for the discussion.
2. The theoretical proof is not sufficient to strongly support the author's point of view, while for the author's so-called new paradigm I suggest to describe it in relation to a specific application scenario.
3. The author's experiments I think are scanty and should be supplemented. Since it is a new paradigm, it should be suitable for all types of tasks and not just limited to deep models for object classification, or it should not be limited to the CV domain, but cover other such as NLP text classification, tabular classification, time-series classification and so on.
4. With the authors' experimental setup, I see that the claimed 100 pre-trained models are different hyper-parameter settings for several commonly used models, such as the number of network layers and so on. This is a different selection of models than what I understand, and I think the authors' experiments are more appropriately classified as either neural architecture search or hyperparameter optimization. Simply by modifying the hyperparameters should not be referred to as a different model.

**Questions:**

1. Can the new paradigm be applied to other areas like NLP, time series analysis, etc.? And provide the corresponding experimental results.
2. The time consumption is not shown in the experimental results, please show the time consumption of each step and compare it with baselines, especially the time for model selection.

---

> ### Author Response · Authors · 2023-11-14
>
> Thank you for your services.
> It is good to know that you find our perspective very interesting.
> Please allow us to provide clarifications regarding your comments.
>
> **1. The author says that he has proposed a new paradigm, but he has only made a little modification in the computation and mapping of embedded vectors, which does not match the innovation of the "new paradigm", and it is more suitable to use the "new method" for the discussion.**
>
> The core of this work is a new paradigm called independently-prepared query-efficient model selection, which indeed only differs from the existing query-efficient paradigm by requiring independent embedding of models. While this change is small, we respectfully argue that it is significant enough to be consider a new paradigm for the following reasons:
>
> >(1) **This small change brings us many benefits**: updatability, decentralizability, flexibility, and certain preservation of candidate privacy, in addition to certain preservation of query privacy inherited from the query-efficient model selection paradigm (as we presented in Section 3.2).
>
> >(2) **None of the existing solutions remain applicable after this small change**: Our solution, standardized embedder, is derived through a very different framework from the existing methods.
>
> We do not know if you are familiar with self-supervised pre-training or not, but it may be a good example to explain our opinions. There is only a small change from the setting of semi-supervised training (i.e. assuming access to both labeled and unlabeled data) to self-supervised learning (i.e. assuming access to only unlabeled data): One simply removes the access to the labeled part. However, the idea of training on labeled data first and using the trained model to provide pseudo-label for unlabeled data is applicable only to semi-supervised settings but not self-supervised settings. We believe self-supervised learning is typically considered a different paradigm from semi-supervised learning.
>
>
> **2. The theoretical proof is not sufficient to strongly support the author's point of view, while for the author's so-called new paradigm I suggest to describe it in relation to a specific application scenario.**
>
> Sorry we are not really sure what your concern is here. Could you please explain a bit more regarding which point of ours is not sufficiently supported? Thank you!
>
> **3. Since it is a new paradigm, it should be suitable for all types of tasks and not just limited to deep models for object classification, or it should not be limited to the CV domain, but cover other such as NLP text classification, tabular classification, time-series classification and so on; Can the new paradigm be applied to other areas like NLP, time series analysis, etc.?**
>
> Notably, standardized embedder is just a proof-of-concept solution to support the feasibility of the new paradigm. While conceptually the new paradigm is generalizable to other domains like language, time series, tabular data and to other tasks like segmentation and detection, we choose to focus on demonstrating its possibility in image classifications, which is arguably one of the most popular settings in previous work studying model selection.
>
> **4. With the authors' experimental setup, I see that the claimed 100 pre-trained models are different hyper-parameter settings for several commonly used models, such as the number of network layers and so on. This is a different selection of models than what I understand, and I think the authors' experiments are more appropriately classified as either neural architecture search or hyperparameter optimization. Simply by modifying the hyperparameters should not be referred to as a different model.**
>
> We respectfully disagree that our evaluated candidate models are obtained ‘simply by modifying the hyperparameters’: While there are indeed some models with similar training recipes and similar architectures (i.e. they are obviously not 100 models with 100 distinct architectures and 100 distinct training recipes), but overall the candidate models we consider offer sufficient diversity regarding architectures (various ConvNets and Transformers) and training recipes (e.g. DINO, Masked Autoencoder, CLIP, Sharpness-aware minimizer). In addition, these models are pre-trained models from torchvision or timm, meaning that they are potentially trained by different teams and many people are actually using them in practice. We argue that while this set is not a perfect representative of every possible use case of model selection, it should be sufficient to support the evaluation of a proof-of-concept solution.

---

> ### Author Response · Authors · 2023-11-14
>
> **5. The time consumption is not shown in the experimental results, please show the time consumption of each step and compare it with baselines, especially the time for model selection.**
>
> Since we did not ensure the consistency of hardwares (e.g. type and occupancy of GPUs) when running experiments, the actual time consumption is not available. However, we provide here the amount of computation required for each stage in our experiments, which can already showcase the scalability of the new paradigm. This information is also available in Section 5.1.
>
> For preparation, we train either 4k steps or 10k steps per candidate model, which is equivalent to about 10 and 25 epochs on the ImageNet validation set (which contains 50k samples). Notably, during preparation, both the candidate model and the baseline features are kept frozen: Only two linear heads and an embedding vector will be updated. These computations are considered quite affordable, especially because each model only needs to be prepared once and different model owners can potentially handle the preparation of their own models. Notably, for the existing method Model2Vec (i.e. query-efficient model selection), each candidate model will also need to be evaluated on multiple predefined downstream tasks before learning all model embeddings jointly, which further limits their usability.
>
> For queries, we use essentially the same training recipes as the final evaluation of downstream accuracy (i.e. linear probing with 60 epochs on the downstream dataset). Approximately and informally, if the amount of computation required to tune and evaluate one model for a given downstream task is $C$, then the primary computations required is $2C$ for standardized embedder and query-efficient model selection, while it will be $O(MC)$ (with some constant) with transferability metrics, where $M$ is the number of candidate models.
>
>
> **Thanks again for your review.
> Please let us know your comments at your earliest convenience.
> Best regards.**

---

> > ### Author Response · Authors · 2023-11-21
> > **A gentle reminder**
> >
> > Dear reviewer SGa4,
> >
> > Thank you again for your services and every minute of your time spent reviewing our paper.
> >
> > With the discussion stage ending in 2 days, we kindly request your feedback regarding whether our response adequately addresses your concerns.
> >
> > Looking forward to your comment!
> >
> > ---
> >
> > Best wishes,
> >
> > Authors of submission 252

---

> > > ### Author Response · Authors · 2023-11-23
> > > **Kind Reminder: Review Deadline Approaching**
> > >
> > > Dear reviewer SGa4,
> > >
> > > As the review deadline approaches, with just less than two hours remaining, we wish to highlight our updates briefly. We've provided more clarification on the focus of the paper, suitable tasks, complex experimental setup of 100 models, and computation consumption.
> > >
> > > We hope these updates facilitate your re-evaluation. Your insights are invaluable to us, and we deeply appreciate your time and attention.
> > >
> > > Warmest regards,
> > >
> > > Authors of submission 252

---

### Author Response · Authors · 2023-11-15
**Prompt for Discussion**

Dear reviewers,

Once again, we extend our sincere gratitude for your dedicated review services.

Your active participation in the ongoing discussion will be crucial to ensuring well-informed decisions.

*Could you please share your comments for rebuttals at your earliest convenience?*

Best regards,

The authors of submission 252